# Dynamics of a Quantum Common-Pool Resource Game with Homogeneous Players’ Expectations

**DOI:** 10.3390/e25121585

**Published:** 2023-11-25

**Authors:** Juan Grau-Climent, Luis García-Pérez, Ramon Alonso-Sanz, Juan Carlos Losada

**Affiliations:** Complex Systems Group, Universidad Politécnica de Madrid, 28040 Madrid, Spain; juan.grau.climent@alumnos.upm.es (J.G.-C.); luis.gperez@alumnos.upm.es (L.G.-P.); ramon.alonso@upm.es (R.A.-S.)

**Keywords:** quantum games, common-pool resource game, entanglement, homogeneous players, local stability analysis, bifurcation and chaos

## Abstract

In this work, we analyse a common-pool resource game with homogeneous players (both have boundedly rational expectations) and entanglement between players’ strategies. The quantum model with homogeneous expectations is a differential approach to the game since, to the best of our knowledge, it has hardly been considered in previous works. The game is represented using a Cournot type payoff functions, limited to the maximum capacity of the resource. The behaviour of the dynamics is studied considering how the fixed points (particularly the Nash equilibrium) and the stability of the system vary depending on the different values of the parameters involved in the model. In the analysis of this game, it is especially relevant to consider the extent to which the resource is exploited, since the output of the players is highly affected by this issue. It is studied in which cases the resource can be overexploited, adjusting the parameters of the model to avoid this scenario when it is possible. The results are obtained from an analytical point of view and also graphically using bifurcation diagrams to show the behaviour of the dynamics.

## 1. Introduction

The first reference to the overuse of common goods was made in 1832 by W. Foster Lloyd in [1]. In this work, he exposed that the point of saturation of an individual resource and a common one shared by a group of individuals is different, which is illustrated with the well-known example of the cattle and the pasture. When you put more cattle on a common pasture, there will be a moment in which the cattle are not properly fed, and then, it is not worth it in terms of benefit. This is only an example to approach to the problem of population growth, which is the real problem to be address. Later, the economist H. Scott Gordon published their article [2] in 1954, in which the economic theory of natural resource utilisation is applied to the fishing industry. Allocating individual fishing quotas (IFQs) or individual transferable quotas (ITQs) is the proposal to solve the problem of fishery.

After these precedent works, G. Hardin wrote their influential article [3] in 1968, where the term ‘tragedy of the commons’ is used as a metaphor which illustrates the argument that free access and unrestricted demand for a finite resource reduces it through overexploitation. In this context, the figure of the free-rider emerges as an individual who uses a common good without contributing to its maintainability, harming the rest of the users of the resource. Hardin focuses on the problem of overpopulation, and from their point of view, ‘freedom to breed will bring ruin to all’. He considers another example of the concept of the use of common resources, as the case of pollution, where the problem is not actually taking something out of the commons but putting something in (for example, noxious and dangerous fumes into the air). Hardin suggests some proposals to avoid the tragedy as selling the resources off as private property or keeping them as a public property but allocating the right to use them.

Elinor Ostrom, winner of the Nobel Prize in 2009 for her contribution to this issue, analysed in [4,5] possible alternatives to avoid the tragedy of the commons in contrast to Hardin’s pessimistic view of human nature. Her work proposes solutions to govern the common resources, exposing that sometimes the problem is solved by voluntary organisations rather than by a coercive state.

On this basis, a lot of relevant research has been developed to analyse the management of common pool resources, many times from the point of view of game theory and, in this context, the common-pool resource game (CPRG) is frequently the term used to refer to it. Most of these studies have modelled the CPRG as a discrete game, typically the prisoner’s dilemma game (PDG), i.e., the disjunctive between cooperation or defection, which means, in this case, the choice of making a good use of the common resource or acting by self-interest, as a free rider. Some examples of works, which follow this line of research are [6,7,8]. Following the model adopted, the majority of the authors have preferred the perspective of players contributing to a common resource instead of consuming it, covered by the PDG including continuous value versions, e.g., [9,10,11,12].

In contrast to the majority of the previous references based on the contribution view of the game, in this article, the dynamics of CPRG is analysed from the perspective of a continuous game where two players consume a common resource. In terms of the model, the functions of a Cournot-type payoff are used (differing from the commonly applied PDG model), limiting the total intake of the resource to its maximum capacity. Additional premises were introduced to enrich the model and in the aim of improving the results obtained, compared to the game with no changes.

The first change in the model consists of including “the entanglement” between the actors in the sense that their behaviours interact and cannot be considered independently. As this entanglement is modelled using quantum methods, the term quantum game theory is commonly used to refer to the study of this type of game. There are some recent reviews of the fundamentals of quantum game theory [13,14], which are interesting to have a general view of this issue. It is relevant to outline that this technique of quantum entanglement between players was used for the first time in [15] showing that, in a zero-sum strategic game, a player can increase their expected payoff by implementing a quantum strategy instead of a classic one. The authors have previously worked with quantum games considering the Cournot game in [16,17], and it has been proven that the quantum game can produce better results in terms of output and stability than the classic game, without entanglement. Concretely, in the CPRG modelling, we use the Li–Du–Massar quantisation scheme proposed in [18], finding that the two players of the Cournot duopoly cooperate in a certain way due to the entanglement between them, since the quantity produced by each firm depends not only on the strategy of that player, but also on the strategy of the other one. This cooperation enables profits to increase as the entanglement is higher. In addition to the previous papers of the authors considering the Cournot game, there are examples in the literature which successfully support the use of the Li–Du–Massar technique in other games such as the Bertrand duopoly game in [19] or the Stackelberg duopoly game in [20]. Other quantisation methods can successfully improve the results of classic game theory, such as the Frackiewicz model proposed in [21]. An interesting example is that of [22], which applies the EWL quantisation protocol in the Frackiewicz–Pykacz parametrisation to the prisoner’s dilemma game, the battle of the sexes, and two versions of the chicken game, showing that, considering this parametrisation in mixed strategies, the Nash equilibria is much closer to the Pareto-efficient solutions than the equilibria of classical games.

Another aspect to take into consideration in the model is whether the players follow the same strategy (homogeneous expectations) as in [23,24,25,26,27], or different strategies (heterogeneous expectations), as in [28,29,30,31]. Generally, three types of players are considered depending on their expectations and the mechanism used to estimate their profits: the boundedly rational player, the adaptive player, and the naive player. Each type of player adjusts the production for future periods to maximise their profit using a different strategy. Our article is inspired by [32], which builds a nonlinear dynamic system of the tragedy of the commons with quantum strategies in discrete time, considering a quadratic uptake’s average function and heterogeneous players. On the contrary, our model implements a lineal uptake’s average function and the case of homogeneous expectations, which can be understood as if both players compete under the same conditions, involving in this case two boundedly rational players. In contrast, to consider this rational behaviour, there is another type of game whose players’ behaviour is inherited and, from there, evolutionary game theory arises. The most notable article on that issue is [33], which studies the ’evolution of social norms’ in specific economic scenarios.

The homogeneous model implies that both players approach the game with exactly the same opportunities to obtain the highest payoff, without there being any advantage for either of them. This is one of the main motivations to follow this type of model, as well as distinguishing our work from other papers with an innovative perspective. In this sense, it is significantly relevant to note that, to the best of our knowledge, the quantum game with homogeneous players is not tackled, except in one of our tragedy works previously cited [17]. Thus, our analysis can be considered a pioneer of its kind.

In this context, the present article is developed in the aim of studying the effect of entanglement in the dynamics of a CPRG with homogeneous players in terms of the variation of the fixed points and the stability considering the different values of the parameters involved in the game.

This paper is organised as follows. In Section 2, a dynamical CPRG with entanglement and homogeneous players is described. The equilibrium points, the local stability condition, and the level of saturation of the common resource are studied in Section 3. In Section 4, we will study the case of the classic game to see the differences and similarities with the quantum game. In Section 5, numerical simulation is used to demonstrate the complicated dynamics of the system and to illustrate the results obtained in the previous section. Section 6 presents the conclusions and meaning of this paper.

## 2. The Game

Based on what is described by G. Hardin [3], we consider two users (in Hardin’s example, it would be two herders raising sheep) which consume a common-pool resource (or pasture) of limited capacity $G_{max}$. These users choose the quantity of uptake $g_{i}$ (number of sheep) and the total uptake *G* (total number of sheep) is given by:(1)$$G=g_{1}+g_{2}.$$ We also have a cost c(c>0) of each uptake (in the case of Hardin’s example, it is the cost of a sheep): (2)$$C_{i}\left(g_{i}\right)=c_{i}g_{i},$$
where $c_{i}$ is the marginal cost of the *i* user, and $c_{i}$ is a positive constant.

$G_{max}$ denotes the maximum amount of uptake allowed in this common-pool resource. We also consider v(G) the marginal (unitary) profit as a function of the total uptake *G*, which decreases as G increases, according to: (3)$$v\left(G\right)=\left\{\begin{array}{cc} v_{0}\left(1-\frac{G}{k\;G_{max}}\right) & \mathrm{if}\;G\leq G_{max}\\ 0 & \mathrm{if}\;G>G_{max} \end{array}\right.,$$
where $v_{0}$ and $G_{max}$ are positive constants and the parameter *k* depicts the degree of elasticity of the model (from the fully elastic game in k = 1.0 up to an inelastic one at k→∞). In this last case, $v\left(G\right)=v_{0}$ and there is no dependency with *G*.

As a result of that, the profits of the two users are given by: (4)$$u_{i}=\left\{\begin{array}{cc} g_{i}\left(v\left(G\right)-c_{i}\right) & \mathrm{if}\;G\leq G_{max}\\ -g_{i}c_{i} & \mathrm{if}\;G>G_{max} \end{array}\right..$$

At this point, we are going to quantise the game according to the Li–Du–Massar entanglement structure based on quantum methods [18,34]. This entangled version has several steps. First, the game starts from the *initial state* |00〉. This state undergoes a *unitary entanglement operation* $\hat{J}\left(\gamma \right)=e^{-\gamma \left({\hat{a}}_{1}^{†}{\hat{a}}_{2}^{†}-{\hat{a}}_{1}{\hat{a}}_{2}\right)}$, where $a_{i}^{†}\left({\hat{a}}_{i}\right)$ represents the creation (annihilation) operator of a player’s *i* electromagnetic field and γ≥0 is known as the *squeezing parameter* and can be reasonably regarded as a measure of entanglement. Next, the two users execute their *strategic moves* via unitary operation ${\hat{D}}_{i}\left(x_{i}\right)=e^{x_{i}\left({\hat{a}}_{i}^{†}-{\hat{a}}_{i}\right)/\sqrt{2}},\;i=1,2$. Finally, the two users’ states are measured after *a disentanglement operation* $\hat{J}{\left(\gamma \right)}^{†}$. Thus, the *final state* is carried out by $|\psi_{f}\rangle =\hat{J}{\left(\gamma \right)}^{†}\left({\hat{D}}_{1}\left(x_{1}\right)⊗{\hat{D}}_{2}\left(x_{2}\right)\right)\hat{J}\left(\gamma \right)|00\rangle$. The final measurement gives the respective quantum uptakes of the two users: (5)$$\begin{array}{c} g_{1}^{c}=g_{1}\cosh\gamma +g_{2}\sinh\gamma ,\\ g_{2}^{c}=g_{2}\cosh\gamma +g_{1}\sinh\gamma , \end{array}$$
where $g_{1}$ and $g_{2}$ represent the independent uptakes and $g_{1}^{c}$ and $g_{2}^{c}$ are the entangled uptakes used by the players in the quantum game. When the degree of entanglement is zero, i.e., γ=0, then the quantum game turns into the original classic form and $g_{i}^{c}=g_{i}$. In the analysis of the game, the negative values of the degree of entanglement are excluded, i.e., we consider γ≥0.

Substituting the Equation (Equation 5) into the Equation (Equation 1), the entangled total quantum uptake is obtained as: (6)$$G^{c}=g_{1}^{c}+g_{2}^{c}=e^{\gamma }\left(g_{1}+g_{2}\right).$$

The profits of the two users are given by Equation (Equation 4). Then, to find the quantum profit, we substitute the Equations (Equation 3), (Equation 5) and (Equation 6) into Equation (Equation 4), resulting: (7)$$\begin{array}{c} u_{1}^{c}=\left\{\begin{array}{cc} \left(g_{1}\cosh\gamma +g_{2}\sinh\gamma \right)\left(v_{0}-\frac{v_{0}}{k\;G_{max}}e^{\gamma }\left(g_{1}+g_{2}\right)-c_{1}\right) & \mathrm{if}\;G^{c}\leq G_{max}\\ -\left(g_{1}\cosh\gamma +g_{2}\sinh\gamma \right)c_{1} & \mathrm{if}\;G^{c}>G_{max} \end{array}\right.,\\ u_{2}^{c}=\left\{\begin{array}{cc} \left(g_{2}\cosh\gamma +g_{1}\sinh\gamma \right)\left(v_{0}-\frac{v_{0}}{k\;G_{max}}e^{\gamma }\left(g_{1}+g_{2}\right)-c_{2}\right) & \mathrm{if}\;G^{c}\leq G_{max}\\ -\left(g_{2}\cosh\gamma +g_{2}\sinh\gamma \right)c_{2} & \mathrm{if}\;G^{c}>G_{max} \end{array}\right.. \end{array}$$

In this work, we study the case of users with similar strategies (homogeneous players), which area both boundedly rational players. A boundedly rational expectations player has their output decision on the basis of the estimation of the marginal profit $\frac{\partial u_{i}}{\partial g_{i}\left(t\right)}$. Then, this boundedly rational player can be denoted as: (8)$$g_{i}\left(t+1\right)=g_{i}\left(t\right)+\alpha_{i}g_{i}\left(t\right)\Phi_{i}\left(t\right)=g_{i}\left(t\right)+\alpha_{i}g_{i}\left(t\right)\frac{\partial u_{i}^{c}}{\partial g_{i}\left(t\right)},$$
where $\alpha_{i}$ is a positive parameter which represents the speed of the adjustment of the i-user.

To continue, we calculate the derivative of $u_{i}\left(t\right)$ with respect to $g_{i}\left(t\right)$ of the Equation (Equation 8): (9)$$\frac{\partial u_{i}^{c}}{\partial g_{i}\left(t\right)}=\left\{\begin{array}{cc} \left(v_{0}-c_{i}\right)\cosh\gamma -\frac{v_{0}}{k\;G_{max}}\left(1+e^{2\gamma }\right)g_{i}\left(t\right)-\frac{v_{0}}{k\;G_{max}}e^{2\gamma }g_{j}\left(t\right) & \mathrm{if}\;G^{c}\leq G_{max}\\ -c_{i}\cosh\gamma & \mathrm{if}\;G^{c}>G_{max} \end{array}\right.,$$
where i=1,2 and j=2,1.

Then, substituting Equation (Equation 9) into Equation (Equation 8), $g_{i}\left(t+1\right)$ is given by:(10)$$g_{i}\left(t+1\right)=\left\{\begin{array}{cc} g_{i}\left(t\right)+\alpha_{i}g_{i}\left(t\right)\left(\left(v_{0}-c_{i}\right)\cosh\gamma -\frac{v_{0}}{k\;G_{max}}\left(1+e^{2\gamma }\right)g_{i}\left(t\right)-\frac{v_{0}}{k\;G_{max}}e^{2\gamma }g_{j}\left(t\right)\right) & \mathrm{if}\;G^{c}\leq G_{max}\\ g_{i}\left(t\right)-\alpha_{i}g_{i}\left(t\right)c_{i}\cosh\gamma & \mathrm{if}\;G^{c}>G_{max} \end{array}\right.,$$
where i=1,2 and j=2,1.

On the basis of Equation (Equation 10), the entangled CPRG with homogeneous players can be described using the following two dimensional discrete time dynamical systems:(11)$$\begin{array}{cc} g_{1}\left(t+1\right) & =\left\{\begin{array}{cc} g_{1}\left(t\right)+\alpha_{1}g_{1}\left(t\right)\left(\left(v_{0}-c_{1}\right)\cosh\gamma -\frac{v_{0}}{k\;G_{max}}\left(1+e^{2\gamma }\right)g_{1}\left(t\right)-\frac{v_{0}}{k\;G_{max}}e^{2\gamma }g_{2}\left(t\right)\right) & \mathrm{if}\;G^{c}\leq G_{max}\\ g_{1}\left(t\right)-\alpha_{1}g_{1}\left(t\right)c_{1}\cosh\gamma & \mathrm{if}\;G^{c}>G_{max} \end{array}\right.,\\ g_{2}\left(t+1\right) & =\left\{\begin{array}{cc} g_{2}\left(t\right)+\alpha_{2}g_{2}\left(t\right)\left(\left(v_{0}-c_{2}\right)\cosh\gamma -\frac{v_{0}}{k\;G_{max}}\left(1+e^{2\gamma }\right)g_{2}\left(t\right)-\frac{v_{0}}{k\;G_{max}}e^{2\gamma }g_{1}\left(t\right)\right) & \mathrm{if}\;G^{c}\leq G_{max}\\ g_{2}\left(t\right)-\alpha_{2}g_{2}\left(t\right)c_{2}\cosh\gamma & \mathrm{if}\;G^{c}>G_{max} \end{array}\right., \end{array}$$ As $g_{i}$ in i=1,2 are the independent uptakes, they must have a positive value and, if at any step, the result of the equation is negative, we consider that this quantity $g_{i}$ is zero.

## 3. Analysis of the Model in the Quantum Game

### 3.1. Equilibrium Points

To find the equilibrium points when $G^{c}\leq G_{max}$ (when $G^{c}>G_{max}$, the equilibrium point is given by $g_{1}=0$ and $g_{2}=0$), we replace all $g_{i}\left(t+1\right)$ and $g_{i}\left(t\right)$ by $g_{i}$ into Equation (Equation 11), obtaining the following nonlinear algebraic system: (12)$$\begin{array}{c} g_{1}\left(\left(v_{0}-c_{1}\right)\cosh\gamma -\frac{v_{0}}{k\;G_{max}}\left(1+e^{2\gamma }\right)g_{1}-\frac{v_{0}}{k\;G_{max}}e^{2\gamma }g_{2}\right)=0,\\ g_{2}\left(\left(v_{0}-c_{2}\right)\cosh\gamma -\frac{v_{0}}{k\;G_{max}}\left(1+e^{2\gamma }\right)g_{2}-\frac{v_{0}}{k\;G_{max}}e^{2\gamma }g_{1}\right)=0. \end{array}$$

In System (Equation 12), there are four fixed points. As can be seen in the equations, one solution is given by $g_{1}=0$ and another by $g_{2}=0$. Then, the first fixed point is $E^{\left(1\right)}=\left(g_{1}^{\left(1\right)},g_{2}^{\left(1\right)}\right)=\left(0,0\right)$.

The second fixed point is given by $g_{1}=0$ in the first equation of Equation (Equation 12) and then, substituting this value into the second equation to obtain $g_{2}$: (13)$$g_{2}=\frac{G_{max}\;k\;\left(v_{0}-c_{2}\right)\cosh\gamma }{v_{0}\left(1+e^{2\gamma }\right)},$$
therefore, this second fixed point is $E^{\left(2\right)}=\left(g_{1}^{\left(2\right)},g_{2}^{\left(2\right)}\right)=\left(0,\frac{G_{max}\;k\;\left(v_{0}-c_{2}\right)\cosh\gamma }{v_{0}\left(1+e^{2\gamma }\right)}\right)$.

The third fixed point is obtained in the same manner but, in this case, considering $g_{2}=0$ in the second equation of the Equation (Equation 12) and then substituting this value into the first equation to obtain $g_{1}$: (14)$$g_{1}=\frac{G_{max}\;k\;\left(v_{0}-c_{1}\right)\cosh\gamma }{v_{0}\left(1+e^{2\gamma }\right)},$$
therefore, this third fixed point is $E^{\left(3\right)}=\left(g_{1}^{\left(3\right)},g_{2}^{\left(3\right)}\right)=\left(\frac{G_{max}\;k\;\left(v_{0}-c_{1}\right)\cosh\gamma }{v_{0}\left(1+e^{2\gamma }\right)},0\right)$.

These points, which represent the uptakes, must be positive and therefore, it must be verified that: (15)$$\begin{array}{c} v_{0}>c_{2},\\ v_{0}>c_{1}. \end{array}$$

The last fixed point is determined by the following equations: (16)$$\begin{array}{c} \left(v_{0}-c_{1}\right)\cosh\gamma -\frac{v_{0}}{k\;G_{max}}\left(1+e^{2\gamma }\right)g_{1}-\frac{v_{0}}{k\;G_{max}}e^{2\gamma }g_{2}=0,\\ \left(v_{0}-c_{2}\right)\cosh\gamma -\frac{v_{0}}{k\;G_{max}}\left(1+e^{2\gamma }\right)g_{2}-\frac{v_{0}}{k\;G_{max}}e^{2\gamma }g_{1}=0. \end{array}$$

Thus, the last fixed point of the system is $E^{\left(4\right)}=\left(g_{1}^{\left(4\right)},g_{2}^{\left(4\right)}\right)$, where:(17)$$g_{1}^{\left(4\right)}=\frac{k\;G_{max}\left(v_{0}-c_{1}+\left(c_{2}-c_{1}\right)e^{2\gamma }\right)\cosh\gamma }{v_{0}\left(1+2e^{2\gamma }\right)},g_{2}^{\left(4\right)}=\frac{k\;G_{max}\left(v_{0}-c_{2}+\left(c_{1}-c_{2}\right)e^{2\gamma }\right)\cosh\gamma }{v_{0}\left(1+2e^{2\gamma }\right)}.$$

Since the fixed point $E^{\left(4\right)}$ represents the uptake of each user, it must be positive and this requires
(18)$$\begin{array}{cc} v_{0}-c_{1}+e^{2\gamma }\left(c_{2}-c_{1}\right) & >0,\\ v_{0}-c_{2}+e^{2\gamma }\left(c_{1}-c_{2}\right) & >0. \end{array}$$

### 3.2. Local Stability

The next step it is the analysis of the stability of the dynamical system, where the Jacobian matrix is given by Equation (Equation 11). Therefore, the Jacobian matrix can be written as:(19)$$J\left(\omega_{1},\omega_{2}\right)=\left(\begin{array}{cc} J_{11} & J_{12}\\ J_{21} & J_{22} \end{array}\right),$$
where the elements of the matrix are given by the following expressions:(20)$$\begin{array}{cc} J_{11} & =1+\alpha_{1}\left[\left(v_{0}-c_{1}\right)\cosh\gamma -\frac{v_{0}}{k\;G_{max}}\left(2\left(1+e^{2\gamma }\right)g_{1}+e^{2\gamma }g_{2}\right)\right],\\ J_{12} & =-\alpha_{1}\frac{v_{0}}{k\;G_{max}}e^{2\gamma }g_{1},\\ J_{21} & =-\alpha_{2}\frac{v_{0}}{k\;G_{max}}e^{2\gamma }g_{2},\\ J_{22} & =1+\alpha_{2}\left[\left(v_{0}-c_{2}\right)\cosh\gamma -\frac{v_{0}}{k\;G_{max}}\left(2\left(1+e^{2\gamma }\right)g_{2}+e^{2\gamma }g_{1}\right)\right]. \end{array}$$

**Theorem** **1.**
*The quantum equilibrium points $E^{\left(1\right)}=\left(g_{1}^{\left(1\right)},g_{2}^{\left(1\right)}\right)=\left(0,0\right)$, $E^{\left(2\right)}=\left(g_{1}^{\left(2\right)},g_{2}^{\left(2\right)}\right)=\left(0,\frac{\left(v_{0}-c_{2}\right)\;kG_{max}\cosh\gamma }{v_{0}\left(1+e^{2\gamma }\right)}\right)$ and $E^{\left(3\right)}=\left(g_{1}^{\left(3\right)},g_{2}^{\left(3\right)}\right)=\left(\frac{\left(v_{0}-c_{1}\right)\;kG_{max}\cosh\gamma }{v_{0}\left(1+e^{2\gamma }\right)},0\right)$ of the dynamical system given by Equation (Equation 11) are unstable.*


**Proof.** Proven in Appendix A. □

**Theorem** **2.**
*The quantum Nash equilibrium point*

*$E^{\left(4\right)}=\left(g_{1}^{\left(4\right)},g_{2}^{\left(4\right)}\right)=\left(\frac{\left(v_{0}-c_{1}+\left(c_{2}-c_{1}\right)e^{2\gamma }\right)\;kG_{max}\cosh\gamma }{v_{0}\left(1+2e^{2\gamma }\right)},\frac{\left(v_{0}-c_{2}+\left(c_{1}-c_{2}\right)e^{2\gamma }\right)\;kG_{max}\cosh\gamma }{v_{0}\left(1+2e^{2\gamma }\right)}\right)$ of the dynamical system given by Equation (Equation 11) is stable provided that*

(21)
$$\begin{array}{cc} \left(a\right) & \alpha_{1}\alpha_{2}{\left(\cosh\gamma \right)}^{2}AB-2\cosh\gamma \left(1+e^{2\gamma }\right)\left(\alpha_{1}A+\alpha_{2}B\right)+4\left(1+2e^{2\gamma }\right)>0,\\ \left(b\right) & \alpha_{1}\alpha_{2}\cosh\gamma AB-\left(1+e^{2\gamma }\right)\left(\alpha_{1}A+\alpha_{2}B\right)<0, \end{array}$$

*where $A=\left[v_{0}-c_{1}+\left(c_{2}-c_{1}\right)e^{2\gamma }\right]$ and $B=\left[v_{0}-c_{2}+\left(c_{1}-c_{2}\right)e^{2\gamma }\right]$ and there is no dependency neither on k nor on $G_{max}$.*


**Proof.** Proven in Appendix B. □

In this paper, the particular case of $\alpha_{1}=\alpha_{2}=\alpha$ is analysed. Considering this condition, Equation (Equation 21) can be written as follows: (22)$$\begin{array}{cc} \left(a\right) & \alpha^{2}{\left(\cosh\gamma \right)}^{2}AB-2\alpha \cosh\gamma \left(1+e^{2\gamma }\right)\left(A+B\right)+4\left(1+2e^{2\gamma }\right)>0,\\ \left(b\right) & \alpha <\frac{\left(1+e^{2\gamma }\right)\left(A+B\right)}{\cosh\gamma AB}. \end{array}$$

In the next proposition, it is shown that these two conditions can be simplified to a unique expression.

**Theorem** **3.**
*In the case $\alpha_{1}=\alpha_{2}=\alpha$, the dynamical system given by Equation (Equation 11) is stable provided that*

(23)
$$\alpha <\frac{\left(1+e^{2\gamma }\right)\left[2v_{0}-\left(c_{1}+c_{2}\right)\right]-\sqrt{e^{4\gamma }{\left[2v_{0}-\left(c_{1}+c_{2}\right)\right]}^{2}+{\left(1+2e^{2\gamma }\right)}^{3}{\left(c_{1}-c_{2}\right)}^{2}}}{\left[v_{0}-c_{1}+\left(c_{2}-c_{1}\right)e^{2\gamma }\right]\left[v_{0}-c_{2}+\left(c_{1}-c_{2}\right)e^{2\gamma }\right]\cosh\gamma }.$$



**Proof.** Proven in Appendix C. □

It can be shown that stability always decreases as α increases in γ≥0 in both cases, namely symmetric ($c_{1}=c_{2}=c$) and asymmetric games ($c_{1}\neq c_{2}).$

Firstly, the symmetric game where $c_{1}=c_{2}=c$ is analysed. In this case, Equation (Equation 23) can be simplified as follows:(24)$$\alpha <\frac{2}{\left(v_{0}-c\right)\cosh\gamma }.$$

The derivative of the expression $g\left(\gamma \right)=\frac{2}{\left(v_{0}-c\right)\cosh\gamma }$ is given by:(25)$$\frac{\partial g\left(\gamma \right)}{\partial \gamma }=\frac{-2\left(v_{0}-c\right)\sinh\gamma }{{\left[\left(v_{0}-c\right)\cosh\gamma \right]}^{2}}.$$

This equation is equal to zero when γ=0 and it always verifies that it is negative in γ>0, which means that stability is a decreasing function of the entanglement degree in γ≥0. If we again derive the previous expression, we obtain:(26)$$\frac{\partial g{\left(\gamma \right)}^{2}}{\partial \gamma^{2}}=\frac{-2\left[\cosh\gamma^{2}-2\sinh\gamma^{2}\right]}{\left(v_{0}-c\right)\cosh\gamma^{2}}.$$

Since the previous expression is negative in γ=0, there is a maximum in this point. This occurs from a mathematical point of view because the negative values of γ are not taken into consideration in this analysis.

In the asymmetric case, following a similar analysis, the same conclusion could be reached, i.e., the stability always decreases as α increases in γ≥0. This result will be graphically shown subsequently in this paper.

### 3.3. Exploitation of the Common-Pool Resource

In this subsection, the extent to which the common-pool resource is exploited is analysed, i.e., the game is studied to obtain the conditions in which the maximum capacity of the good is not exceeded.

The condition to avoid the overexploitation of the resource can be expressed as follows:(27)$$G^{c}=g_{1}^{c}+g_{2}^{c}=e^{\gamma }\left(g_{1}+g_{2}\right)=\frac{\left(2v_{0}-\left(c_{1}+c_{2}\right)\right)\left(e^{2\gamma }+1\right)}{2v_{0}\left(1+2e^{2\gamma }\right)}kG_{max}\leq G_{max}$$ On the basis of this condition, the range of values of *k* and γ in which the capacity of the resource is not exceeded is analysed.

**Theorem** **4.**
*Three different regions depending on the value of k must be considered to analyse the behaviour of the game in terms of the exploitation of the common resource:*

*1.* 
*$k\leq k^{•}$: the exploitation of the resource is always under its maximum capacity.*
*2.* 
*$k^{•}<k<k_{max}$: the level of exploitation of the resource depends on the value of γ:*
*(a)* 
*$\gamma \leq \gamma^{•}$: the limit of the resource is exceeded.*
*(b)* 
*$\gamma >\gamma^{•}$: the limit of the resource is not exceeded.*
*3.* 
*$k\geq k_{max}$: the exploitation of the resource is always over its maximum capacity.*


*The values of $k_{max}$, $k^{•}$ and $\gamma^{•}$ are given by:*

(28)
$$k_{max}=\frac{4v_{0}}{2v_{0}-\left(c_{1}+c_{2}\right)}$$


(29)
$$k^{•}=\frac{2v_{0}\left(1+2e^{2\gamma }\right)}{\left(2v_{0}-\left(c_{1}+c_{2}\right)\right)\left(e^{2\gamma }+1\right)},$$


(30)
$$\gamma^{•}=\frac{1}{2}\ln\frac{2v_{0}\left(1-k\right)+\left(c_{1}+c_{2}\right)k}{2v_{0}\left(k-2\right)-\left(c_{1}+c_{2}\right)k}.$$



**Proof.** Proven in Appendix D. □

## 4. Analysis of the Model in Classic Game

### 4.1. The Game

We will study the case of the classic game to see the differences and similarities with the quantum game. Considering this case, the profits of the two users are given by Equation (Equation 7) without entanglement: (31)$$\begin{array}{c} u_{1}=\left\{\begin{array}{cc} g_{1}\left(v_{0}-\frac{v_{0}}{k\;G_{max}}\left(g_{1}+g_{2}\right)-c_{1}\right) & \mathrm{if}\;G\leq G_{max}\\ -g_{1}c_{1} & \mathrm{if}\;G>G_{max} \end{array}\right.,\\ u_{2}=\left\{\begin{array}{cc} g_{2}\left(v_{0}-\frac{v_{0}}{k\;G_{max}}\left(g_{1}+g_{2}\right)-c_{2}\right) & \mathrm{if}\;G\leq G_{max}\\ -g_{2}c_{2} & \mathrm{if}\;G>G_{max} \end{array}\right.. \end{array}$$ Following the homogeneous expectations model previously exposed, the dynamical system can be written: (32)$$g_{i}\left(t+1\right)=g_{i}\left(t\right)+\alpha_{i}g_{i}\left(t\right)\frac{\partial u_{i}^{c}}{\partial g_{i}\left(t\right)}=\left\{\begin{array}{cc} g_{i}\left(t\right)+\alpha_{i}g_{i}\left(t\right)\left(v_{0}-c_{i}-\frac{2v_{0}}{k\;G_{max}}g_{i}\left(t\right)-\frac{v_{0}}{k\;G_{max}}g_{j}\left(t\right)\right) & \mathrm{if}\;G\leq G_{max}\\ g_{i}\left(t\right)-\alpha_{i}g_{i}\left(t\right)c_{i} & \mathrm{if}\;G>G_{max} \end{array}\right..$$

### 4.2. Equilibrium Point

Following the same steps as in Section 3.1, the four fixed points in the classic game are: (33)$$\begin{array}{cc} E^{\left(1\right)} & =\left(g_{1}^{\left(1\right)},g_{2}^{\left(1\right)}\right)=\left(0,0\right),\\ E^{\left(2\right)} & =\left(g_{1}^{\left(2\right)},g_{2}^{\left(2\right)}\right)=\left(0,\frac{G_{max}\;k\;\left(v_{0}-c_{2}\right)}{2v_{0}}\right),\\ E^{\left(3\right)} & =\left(g_{1}^{\left(3\right)},g_{2}^{\left(3\right)}\right)=\left(\frac{G_{max}\;k\;\left(v_{0}-c_{1}\right)}{2v_{0}}\right),\\ E^{\left(4\right)} & =\left(g_{1}^{\left(4\right)},g_{2}^{\left(4\right)}\right)=\left(\frac{\left(v_{0}-2c_{1}+c_{2}\right)\;kG_{max}}{3v_{0}},\frac{\left(v_{0}-2c_{2}+c_{1}\right)\;kG_{max}}{3v_{0}}\right). \end{array}$$ As the equilibrium points must be positive, it must be verified: (34)$$\begin{array}{cc} v_{0}-c_{1} & >0,\\ v_{0}-c_{2} & >0,\\ v_{0}-2c_{1}+c_{2} & >0,\\ v_{0}-2c_{2}+c_{1} & >0. \end{array}$$

### 4.3. Local Stability

Considering the particular case of the classic game, we can state the corollaries of the theorems exposed in Section 3.2:

**Corollary** **1.**
*The quantum equilibrium points $E^{\left(1\right)}=\left(g_{1}^{\left(1\right)},g_{2}^{\left(1\right)}\right)=\left(0,0\right)$, $E^{\left(2\right)}=\left(g_{1}^{\left(2\right)},g_{2}^{\left(2\right)}\right)=\left(0,\frac{\left(v_{0}-c_{2}\right)\;kG_{max}}{2v_{0}}\right)$ and $E^{\left(3\right)}=\left(g_{1}^{\left(3\right)},g_{2}^{\left(3\right)}\right)=\left(\frac{\left(v_{0}-c_{1}\right)\;kG_{max}}{2v_{0}},0\right)$ of the dynamical system given by Equation (Equation 32) are unstable.*


**Proof.** Following the same steps as in Appendix A but with γ=0, at least one of the eigenvalues verifies: $\left|\lambda_{i}\right|>1.$ □

**Corollary** **2.***The quantum Nash equilibrium point* 
*$E^{\left(4\right)}=\left(g_{1}^{\left(4\right)},g_{2}^{\left(4\right)}\right)=\left(\frac{\left(v_{0}-2c_{1}+c_{2}\right)\;kG_{max}}{3v_{0}},\frac{\left(v_{0}-2c_{2}+c_{1}\right)\;kG_{max}}{3v_{0}}\right)$ of the dynamical system given by Equation (Equation 32) is stable provided that:*
(35)$$\begin{array}{cc} \left(a\right) & \alpha_{1}\alpha_{2}AB-4\left(\alpha_{1}A+\alpha_{2}B\right)+12>0,\\ \left(b\right) & \alpha_{1}\alpha_{2}AB-2\left(\alpha_{1}A+\alpha_{2}B\right)<0, \end{array}$$
*where $A=v_{0}-2c_{1}+c_{2}$ and $B=v_{0}-2c_{2}+c_{1}$ and that there is no dependency neither on k nor on $G_{max}$.*

**Proof.** Following the same steps as in Appendix B but with γ=0. □

**Corollary** **3.**
*In the case where $\alpha_{1}=\alpha_{2}=\alpha$, the dynamical system given by Equation (Equation 32) is stable provided that*

(36)
$$\alpha <\frac{2\left(2v_{0}-\left(c_{1}+c_{2}\right)\right)-\sqrt{{\left(2v_{0}-\left(c_{1}+c_{2}\right)\right)}^{2}+27{\left(c_{1}-c_{2}\right)}^{2}}}{\left(v_{0}-2c_{1}+c_{2}\right)\left(v_{0}-2c_{2}+c_{1}\right)}.$$



**Proof.** Following the same steps as in Appendix C but with γ=0. □

**Corollary** **4.**
*In this case, two different regions depending on the value of k must be considered to analyse the behaviour of the game in terms of the exploitation of the common resource:*

*1.* 
*$k\leq k^{•}$: the exploitation of the resource is always under its maximum capacity.*
*2.* 
*$k^{•}<k$: the exploitation of the resource is always over its maximum capacity.*


*The value of $k^{•}$ is given by:*

(37)
$$k^{•}=\frac{3v_{0}}{2v_{0}-\left(c_{1}+c_{2}\right)},$$



**Proof.** Following the same steps as in Appendix D but with γ=0. □

## 5. Numerical Simulation

In this section, the numerical simulation is used to observe the variation of the stability and the dynamical behaviour of the system, supporting the results analytically obtained throughout this article. Two cases are included in this study, namely the symmetric and asymmetric games (i.e., with equal and different costs) to show the differences between them.

At first, Figure 1 represents the variation of the stability region as a function of the degree of entanglement, γ, with an example of each type of game. Then, Figure 1a shows the symmetric game considering the following values: $v_{0}=4,\;c_{1}=0,\;c_{2}=0$. Finally, Figure 1b represents the asymmetric game setting the following values: $v_{0}=4,\;c_{1}=0,\;c_{2}=1$. In all cases, it is shown that the stability region decreases as the degree of entanglement increases.

Hereafter, the behaviour of the two-dimensional discrete time dynamical system given by Equation (Equation 11) is represented in Figure 2 and Figure 3. These figures consider the different values of some specific parameters, γ and *k*, to graphically study the dynamics in terms of the stability and exploitation of the common-pool resource, showing: the bifurcation diagrams of the output of the two firms ($g_{1}$ and $g_{2}$), $G=g_{1}+g_{2}$, and $G^{c}=g_{1}^{c}+g_{2}^{c}$ (level of exploitation of the common-pool resource) and the constant $G_{max}=12$ (maximum capacity of the common-pool resource).

Firstly, the symmetric game is tackled, considering the following values of the parameters: $v_{0}=4,c_{1}=0,c_{2}=0,G_{max}=12$ in Figure 2. Since the game is perfectly symmetrical, it is observed that the uptakes of both players are equal. They overlap in the figures and are shown in red since $g_{2}$ (red) is represented after $g_{1}$ (blue). The following figures show several bifurcation diagrams in which, depending on the value of *k* and γ, the behaviour of the game is different:γ=0.2, k=1.2: in this case, $k^{•}=1.5987$ and, since $k=1.2<k^{•}$, the exploitation of the common resource is under its maximum capacity (as there must be ∀γ≥0).γ=0.2, k=1.7: compared to the previous case, the value of *k* increases and verifies $k=1.7>k^{•}$. As $\gamma <\gamma^{•}=0.4236$, therefore, the resource is overexploited. Stability is preserved since γ do not vary.γ=0.7, k=1.7: the value of γ increases above $\gamma^{•}=0.4236$, limiting the exploitation of the resource under its maximum. In this case, stability decreases as γ increases compared to the previous figure.γ=0.7, k=2.3: the value of *k* is above $k_{max}=2$, and therefore, the maximum capacity of the resource is exceeded (as there must be ∀γ≥0). Stability is preserved compared to the previous case, because the value of γ does not vary.

It is relevant to outline that the previous analysis applies to the stability zone, but the behaviour of the game can be different in the chaos zone. For example, in Figure 2c, the resource is not overexploited in the stability zone, but the maximum capacity is clearly exceeded in the chaos zone. Note that the values of $G_{c}$ over $G_{max}$ are represented in the figures, although this case is not analytically possible, to show the saturation of the common resource and the extent to which it occurs.

Secondly, the asymmetric game is studied, considering the following values of the parameters: $v_{0}=4,\;c_{1}=0,\;c_{2}=1,\;G_{max}=12$ in Figure 3. Since $c_{2}>c_{1}$, the uptake of player 1 is greater than the correspondence to player 2. As it occurs in the symmetric case, depending on the value of *k* and γ, the behaviour of the asymmetric game is different:γ=0.2, k=1.6: in this case, $k^{•}=1.8271$, and since $k=1.6<k^{•}$, the resource it is not exploited beyond its limits (as there must be ∀γ≥0).γ=0.2, k=1.9: the value of *k* increases, compared to the previous value, and it verifies that $k=1.9>k^{•}$. As $\gamma <\gamma^{•}=0.3372$, therefore, the maximum capacity of the resource is exceeded. Stability is preserved since γ does not vary.γ=0.4, k=1.9: the value of γ increases above $\gamma^{•}=0.3372$, limiting the exploitation of the resource under its maximum. In this case, stability decreases as γ increases, compared to the previous figure.γ=0.4, k=2.4: it verifies that $k>k_{max}=2.2857$, and therefore, the resource is overexploited (there must be ∀γ≥0). Stability is preserved, compared to the previous case, because the value of γ does not vary.

The previous analysis is valid in the stability zone, but as it occurs in the symmetric game, the behaviour of the game can be different in the chaos zone. In terms of the exploitation of the resource, Figure 3a,c shows that the resource is always under its maximum in the stability zone as opposed to the behaviour in the chaos zone, where overexploitation emerges. Similarly to the symmetric case, the values of $G_{c}$ over $G_{max}$ are represented in the figures, although this case is not analytically possible, to show the saturation of the common resource and the extent to which it occurs.

### Classic Game

Finally, the classic game is studied for both the symmetric and asymmetric game, considering the following values of the parameters: $v_{0}=4,\;G_{max}=12,\;\gamma =0$, and $c_{1}=0,\;c_{2}=0$ for the symmetric case and $c_{1}=0,\;c_{2}=1$ for the asymmetric case, in Figure 4. In the asymmetric case $c_{2}>c_{1}$, the uptake of player 1 is greater than that corresponding to player 2. Depending on the value of *k*, the behaviour is different in both cases:Symmetric case: $c_{1}=c_{2}=0,\;k=1.0$: in this case, $k^{•}=1.5$, and since $k=1.0<k^{•}$, the resource is not exploited beyond its limits.Symmetric case: $c_{1}=c_{2}=0,\;k=1.7$: the value of *k* increases, compared to the previous value, and it verifies $k=1.7>k^{•}$. Then, the maximum capacity of the resource is exceeded.Asymmetric case: $c_{1}=0,\;c_{2}=1,\;k=1.5$: in this case, $k^{•}=1.71428$ and, since $k=1.5<k^{•}$, the resource it is not exploited beyond its limits.Asymmetric case: $c_{1}=0,\;c_{2}=1,\;k=2$: the value of *k* increases, compared to the previous value, and it verifies that $k=2>k^{•}$. Then, the maximum capacity of the resource is exceeded.

This analysis is valid in the stability zone but the behaviour of the game can be different in the chaos zone. In terms of the exploitation of the resource, Figure 4c shows that the resource is always under its maximum in the stability zone as opposed to the behaviour in the chaos zone, where overexploitation emerges. The values of *G* over $G_{max}$ are represented in the figures, although this case is not possible analytically, to show the saturation of the common resource and the extent to which it occurs.

Comparing both types of games, classic and quantum, we previously mentioned that the stability zone decreases as the entanglement increases, as a potential negative effect. However, entanglement is also an effective mechanism of chaos control, since higher values of the degree of the degree of entanglement enable an increase in the value of the parameter $k^{•}$ (as can be seen in Equation(Equation 29), avoiding the saturation of the common resource. In addition, the payoffs in the quantum game improve as the parameter *k* increases, as shown in Equations (Equation 7) and (Equation 11). Therefore, these two variables can be modulated to avoid the overexploitation of the resource and increase the payoffs, but it is also important to take into account the impact of the degree of entanglement in the stability of the system.

## 6. Conclusions

In this article, we studied the dynamics of a common-pool resource game (CPRG) with entanglement between the homogeneous expectations of the players, both boundedly rational type. The symmetric and asymmetric games are considered, i.e., the games with equal and different marginal costs, respectively.

Under these premises, the variation in the fixed points and the stability of the system were studied, as usual in the dynamics, and additionally, in this case, the level of exploitation of the common resource has been observed. The different values of the parameters involved in the game have been considered to obtain the following results.

Firstly, entanglement aggravates the stability of the system, i.e., the higher the entanglement is, the lower the stability in terms of the speed of adjustment is. Another important result is that the stability zone does not depend on either the parameter k of the CPRG or on Gmax (the limit of the common-pool resource).

We have also come to the conclusion that the level of exploitation or saturation of the common resource can be controlled through the degree of entanglement and the parameter k. It is also a relevant point that, although there was no overexploitation in the stable zone of the system, it may exist in the chaos zone, exceeding the limit of the resource.

The quantum game, compared to the classic game, contributes to the control of the saturation of the common resource and improving the payoffs through the correct modulation of the degree of entanglement and the parameter *k*. As entanglement also has a negative impact on the stability of the system, it is important to balance both effects to optimise the system.

All these statements are analytically proven and graphically supported by several local stability figures, and conventional bifurcation diagrams, varying the values of the parameters involved in the analysis to observe the system under different conditions. 

## Figures and Tables

**Figure 1 entropy-25-01585-f001:**
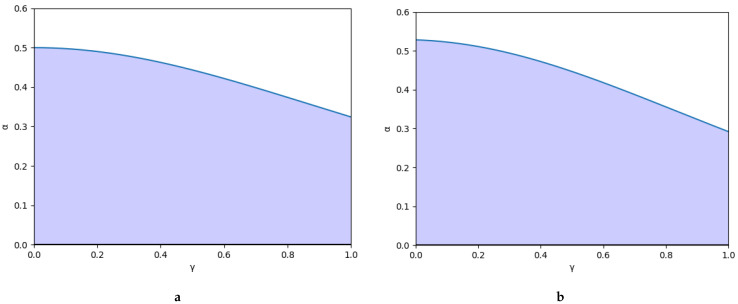
Stability region considering: (**a**) symmetric game ($v_{0}=4,\;c_{1}=0,\;c_{2}=0$); and (**b**) asymmetric game ($v_{0}=4,\;c_{1}=0,\;c_{2}=1$).

**Figure 2 entropy-25-01585-f002:**
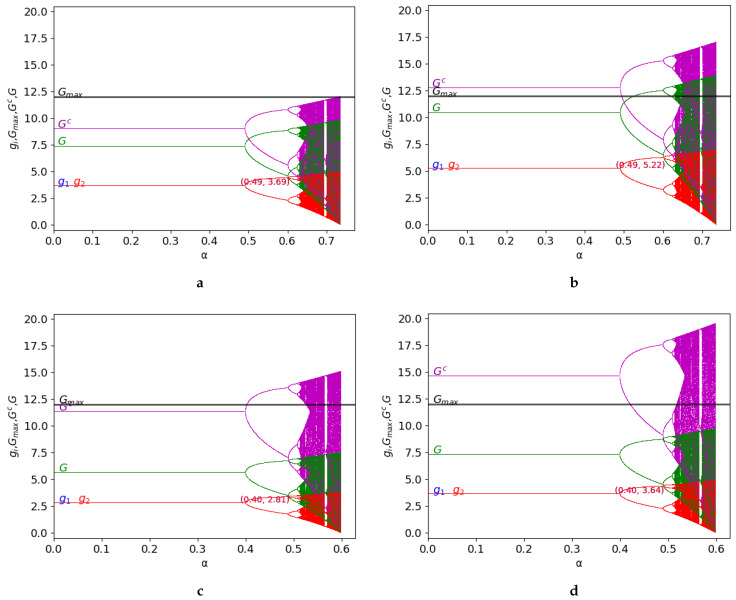
Symmetric game: bifurcation diagrams of the output of the two firms ($g_{1}$ in blue and $g_{2}$ in red), of $G=g_{1}+g_{2}$ (in green), and of $G^{c}=g_{1}^{c}+g_{2}^{c}$ (in purple) as a function of α with $v_{0}=4,c_{1}=0,\;c_{2}=0$. The constant $G_{max}=12$ represents the maximum capacity of the resource (in black). The figure is shown for different values of γ and *k* ($k^{•}=1,5987$ in γ=0.2, $\gamma^{•}=0.4236$ in k=1.7, $k_{max}=2$): (**a**) γ=0.2, k=1.2; (**b**) γ=0.2, k=1.7; (**c**) γ=0.7, k=1.7, and (**d**) γ=0.7, k=2.2.

**Figure 3 entropy-25-01585-f003:**
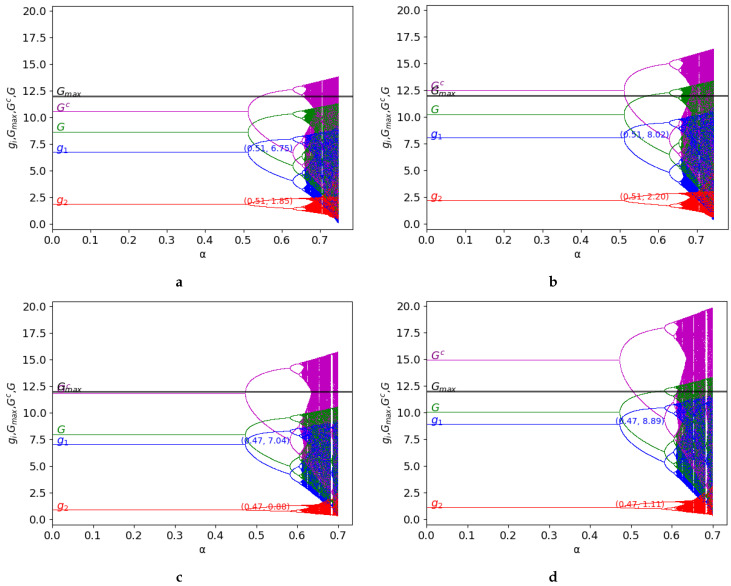
Asymmetric game: bifurcation diagrams of the output of the two firms ($g_{1}$ in blue and $g_{2}$ in red), of $G=g_{1}+g_{2}$ (in green), and of $G^{c}=g_{1}^{c}+g_{2}^{c}$ (in purple) as a function of α with $v_{0}=4,\;c_{1}=0,\;c_{2}=1$. The constant $G_{max}=12$ represents the maximum capacity of the resource (in black). The figure is shown for different values of γ and *k* ($k^{•}=1.8271$ in γ=0.2, $\gamma^{•}=0.3372$ in k=1.9, $k_{max}=2.2857$): (**a**) γ=0.2, k=1.6; (**b**) γ=0.2, k=1.9; (**c**) γ=0.4, k=1.9 and (**d**) γ=0.4, k=2.4.

**Figure 4 entropy-25-01585-f004:**
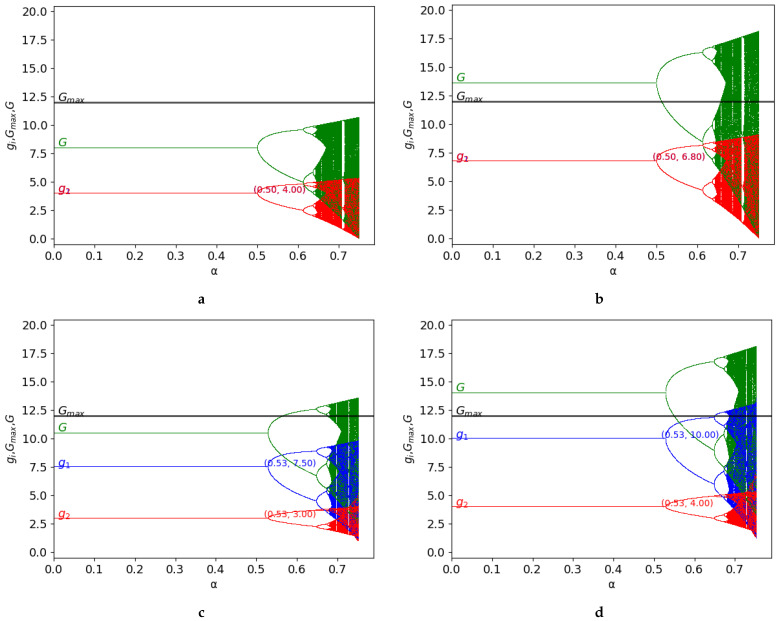
Classic game: the bifurcation diagrams of the output of the two firms ($g_{1}$ in blue and $g_{2}$ in red), of $G=g_{1}+g_{2}$ (in green) as a function of α with $v_{0}=4$. The constant $G_{max}=12$ represents the maximum capacity of the resource (in black). The figure is shown for the different values of *k* in the case of a symmetric game ($c_{1}=0$ and $c_{2}=0$ and therefore $k^{•}=1.5$ in γ=0) and the asymmetric game ($c_{1}=0$ and $c_{2}=1$ and therefore $k^{•}=1.71428$ in γ=0): (**a**) $c_{1}=0$, $c_{2}=0$, γ=0, k=1; (**b**) $c_{1}=0$, $c_{2}=0$, γ=0, k=1.7; (**c**) $c_{1}=0$, $c_{2}=1$, k=1.5; and (**d**) $c_{1}=0$, $c_{2}=1$, k=2.

## Data Availability

The data that support the findings of this study are available within the article.

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
