# Peer review of "Dynamics of a Quantum Common-Pool Resource Game with Homogeneous Players’ Expectations"

_entropy, 2023, doi:10.3390/e25121585_

Round 1

Reviewer 1 Report

Comments and Suggestions for Authors

1.Advantages:

-The topic of studying common-pool resource games using quantum game theory is novel and interesting. Applying quantum entanglement between players' strategies provides new insights into cooperation and stability.

-The authors provide a detailed theoretical analysis of the system dynamics, deriving analytical expressions for equilibrium points, stability conditions, and resource exploitation levels. The propositions and proofs are mathematically rigorous.

-The numerical simulations support the analytical results and illustrate how parameters like entanglement and elasticity affect stability, chaos, and overexploitation of the resource. The bifurcation diagrams clearly show the impact of varying key parameters.

2.Disadvantages:

-The manuscript is very dense and mathematical. More intuitive explanations connecting the equations to real-world meanings would make it more accessible.

-There are no comparisons to the classical version of the game without entanglement. Showing how quantum strategies change outcomes would better highlight the value of using quantum game theory.

-The authors claim this is the first quantum CPR game with homogeneous players, but the uniqueness could be emphasized more strongly.

-The conclusions are brief and reiterate the main results. More insight into practical implications for managing shared resources would be valuable.

3.Suggested Modifications:

-Add more intuitive, plain language explanations of the equations and results. Connect the math clearly to real-world concepts.

-Include analysis of the classical version without entanglement for comparison. Enhance the "value added" from quantum strategies.

-Clearly state in the introduction that this is the first analysis of a quantum CPR game with homogeneous players and the significance of that modeling choice.

-The Discussion needs to be a coherent and cohesive set of arguments that take us beyond this study in particular and help us see the relevance of what the authors have proposed. The authors need to contextualize the findings in the literature and need to be explicit about the added value of your study towards that literature. Also, other studies should be cited to increase the theoretical background of each of the methods used. Findings should be contextualized in the literature and should be explicit about the added value of the study towards the literature. The contribution and implications of the article are yet to be specified. Please refer to the style, DOI: 10.1002/asjc.3199 or 10.1016/j.physa.2020.124993 or 10.1016/j.ejor.2017.07.016 or 10.1016/j.eneco.2023.106827

-Shorten overly detailed derivations in the main text and move additional technical details to appendices or supplementary information if needed.

-Consider adding some references to recent quantum game theory overview papers to reinforce the fundamentals for a broad audience.

Comments on the Quality of English Language

Minor editing of English language required.

Author Response

Reviewer 1

1.Advantages:

-The topic of studying common-pool resource games using quantum game theory is novel and interesting. Applying quantum entanglement between players' strategies provides new insights into cooperation and stability.

-The authors provide a detailed theoretical analysis of the system dynamics, deriving analytical expressions for equilibrium points, stability conditions, and resource exploitation levels. The propositions and proofs are mathematically rigorous.

-The numerical simulations support the analytical results and illustrate how parameters like entanglement and elasticity affect stability, chaos, and overexploitation of the resource. The bifurcation diagrams clearly show the impact of varying key parameters.

2.Disadvantages:

-The manuscript is very dense and mathematical. More intuitive explanations connecting the equations to real-world meanings would make it more accessible.

-There are no comparisons to the classical version of the game without entanglement. Showing how quantum strategies change outcomes would better highlight the value of using quantum game theory.

-The authors claim this is the first quantum CPR game with homogeneous players, but the uniqueness could be emphasized more strongly.

-The conclusions are brief and reiterate the main results. More insight into practical implications for managing shared resources would be valuable.

3.Suggested Modifications:

-Add more intuitive, plain language explanations of the equations and results. Connect the math clearly to real-world concepts.

>> In some equations, we have explained real-world concepts in blue text.

-Include analysis of the classical version without entanglement for comparison. Enhance the "value added" from quantum strategies.

>> Thank you very much for your appreciation. We have added a new section (section 4) to reflect the classic game and be compared to quantum games in section 5. The comparison between the quantum and classic game is also included in the conclusions.

-Clearly state in the introduction that this is the first analysis of a quantum CPR game with homogeneous players and the significance of that modeling choice.

>>It is outlined more emphatically the innovative point of view as well as the meaning of the homogeneous expectations model in the introduction and it is also mentioned in the abstract.

-The Discussion needs to be a coherent and cohesive set of arguments that take us beyond this study in particular and help us see the relevance of what the authors have proposed. The authors need to contextualize the findings in the literature and need to be explicit about the added value of your study towards that literature. Also, other studies should be cited to increase the theoretical background of each of the methods used. Findings should be contextualized in the literature and should be explicit about the added value of the study towards the literature. The contribution and implications of the article are yet to be specified. Please refer to the style, DOI: 10.1002/asjc.3199 or 10.1016/j.physa.2020.124993 or 10.1016/j.ejor.2017.07.016 or 10.1016/j.eneco.2023.106827.

>> The Introduction has been enriched with new references to help the reader to contextualise the added value of this study, following your suggestion. The differences compared to the majority of previous works have been remarked, outlining that our model is based on the Cournot game (not on Prisioner’s Dilemma game), homogeneous expectations (it is pioneer in this sense) and the view of consuming vs. contributing to a common resource. Particularly, we have cited one of the papers you comment as an interesting example of a Cournot duopoly game using the Caputo fractional-order difference calculus: Xin, B., Peng, W.; Kwon, Y. A discrete fractional-order Cournot duopoly game. Physica A: Statistical Mechanics and its Applications 2020, 558, p. 124993. In the part of quantization methods, besides Li-Du-Massar scheme, another relevant technique proposed by Frackiewicz is cited, as well as an example published in Entropy with a complete comparison of classic and quantum game theory considering prisoner’s dilemma, battle of the sexes and the game of chicken (Szopa, Marek. Efficiency of classical and quantum games equilibria. Entropy 2021, 23(5), 506).

-Shorten overly detailed derivations in the main text and move additional technical details to appendices or supplementary information if needed.

>> It has been made according to your consideration. We have added an appendix containing all the theorems. Thank you very much for the appreciation

-Consider adding some references to recent quantum game theory overview papers to reinforce the fundamentals for a broad audience.

>>Two recent references have been added to satisfy the curiosity of potential readers interested in approaching to the quantum game theory from a global point of view. In our opinion, both of them contain a review of the basis of quantum games: Avishai, Yshai. On Topics in Quantum Games. Journal of Quantum Information Science 2023, 13(3), 79–130 and R. Pérez-Antón, J. I. López Sánchez, A. Corbi Bellot. The Game Theory in Quantum Computers: A Review. International Journal of Interactive Multimedia and Artificial Intelligence 2023, In press, 1--9.

Comments on the Quality of English Language

Minor editing of English language required.

>> Thank you very much for your comment. We have reviewed the paper to improve our English.

Reviewer 2 Report

Comments and Suggestions for Authors

The analysis looks to be sound. I would recommend discussing evolutionary stability results with fixed payoffs for comparison. Motivation for the exercise in the paper would also help the reader.

My main comment was that the authors could provide more background on the reasons the work is interesting. They give some general background on common resource games and quantum information theory.
The main results is that entanglement makes the system less stable. They seem to do a responsible job with the analysis. What I don't know is why including entanglement is interesting in this context. Note that there is very little discussion of the interpretation of the results. What they could add is references to other dynamic game theory studies of common resource games such as Sethi and Somanathan (1996), "The Evolution of Social Norms in Common Property Resource Use," American Economic Review v86(4) and explain the relation of their work to the others. The author may not think much about applications, and they are primarily interested in the math.

Author Response

Reviewer 2

The analysis looks to be sound. I would recommend discussing evolutionary stability results with fixed payoffs for comparison. Motivation for the exercise in the paper would also help the reader.

>> The payoffs are given by Eq.(4). Since gi and ci are inherent to the model, the marginal profit v(G) is the only variable which can turn into a constant to fix the payoffs, being then v(G)=v0 (k→∞) and ui=gi(v0-ci). Considering this case, from Eq.(12) with k→∞, the only one fixed point is (g1,g2)=(0,0), which is not stable (proof of stability of (g1,g2)=(0,0)  in Theorem 1 in Appendix A is also valid to this example). The linear decreasing of v(G) as G increases, modulated by the parameter k, avoids this situation.

The main motivation of the paper is to study the level of exploitation of the common resource, avoiding to exceed its capacity. As previous steps, the fixed points, the Nash Equilibrium and the stability of the system are analysed. The control of the overexploitation is achieved by varying the degree of entanglement and the parameter k. Therefore, the inclusion of this variable k in the model discussed in the preceding paragraph plays an important role in the model.

My main comment was that the authors could provide more background on the reasons the work is interesting. They give some general background on common resource games and quantum information theory.
The main result is that entanglement makes the system less stable. They seem to do a responsible job with the analysis. What I don't know is why including entanglement is interesting in this context. Note that there is very little discussion of the interpretation of the results. What they could add is references to other dynamic game theory studies of common resource games such as Sethi and Somanathan (1996), "The Evolution of Social Norms in Common Property Resource Use," American Economic Review v86(4) and explain the relation of their work to the others. The author may not think much about applications, and they are primarily interested in the math.

>> As we have mentioned in the previous point, the main motivation of this work is to study the level of exploitation of the common resource, avoiding exceeding its capacity. To do this, including entanglement makes the system less stable, but it also controls overexploitation. We have referenced the article you have suggested, Sethi, R., Somanathan, E. The Evolution of Social Norms in Common Property Resource Use. The American Economic Review 1996, 66-788. This work is based on evolutionary game theory, i.e. players whose behaviour is inherited, in contrast to our article which is based on a type of rational player.

Reviewer 3 Report

Comments and Suggestions for Authors

In this research, a Common-Pool Resource Game is examined, involving homogeneous players with boundedly rational expectations and interdependence among their strategies. The game is portrayed through Cournot-type payoff functions, constrained by the resource's maximum capacity. The study focuses on understanding the dynamics of the game, specifically how fixed points, notably the Nash Equilibrium, and system stability change in response to varying model parameters. A key aspect of this analysis is the degree of resource exploitation, as player output is significantly impacted by this factor. The research investigates under what conditions resource overexploitation may occur and explores adjustments to model parameters to prevent such a scenario where feasible. The findings are derived through both analytical methods and visual representation via bifurcation diagrams to illustrate the dynamics' behavior.

The paper presents a comprehensive and insightful analysis of a Common-Pool Resource Game with several noteworthy attributes that contribute to its positive perception.

            The research addresses an important real-world issue concerning resource management and exploitation. This relevance is essential in the field of economics, as it can provide valuable insights into strategies for sustainable resource use.

            The paper delves into the dynamics of the game, particularly focusing on the behavior of fixed points, including the Nash Equilibrium, and the stability of the system. This theoretical depth adds to its academic significance.

The incorporation of Cournot-type payoff functions and the study of resource overexploitation provide a bridge between economic theory and environmental concerns. This interdisciplinary approach is both innovative and timely.

The paper employs a combination of analytical and graphical methods, including bifurcation diagrams. This comprehensive approach makes it accessible to a wider audience and enhances the clarity of the findings.

By investigating how parameter adjustments can prevent resource overexploitation, the research offers actionable insights that can inform policy decisions. This practical application contributes to its societal impact.

In summary, the paper offers a rigorous and multifaceted examination of a pressing economic and environmental issue. Its relevance, depth, interdisciplinary approach, analytical methods, and potential policy implications make it a valuable contribution to the field of economic theory and resource management.

I have only to note:

-Regarding equations very often something is missing on the left-hand side in front of the bracket or for what are the brackets good?

-The English can be slightly improved. I propose to proofread the paper again, at best by someone who has good English skills

Comments on the Quality of English Language

Author Response

Reviewer 3

In this research, a Common-Pool Resource Game is examined, involving homogeneous players with boundedly rational expectations and interdependence among their strategies. The game is portrayed through Cournot-type payoff functions, constrained by the resource's maximum capacity. The study focuses on understanding the dynamics of the game, specifically how fixed points, notably the Nash Equilibrium, and system stability change in response to varying model parameters. A key aspect of this analysis is the degree of resource exploitation, as player output is significantly impacted by this factor. The research investigates under what conditions resource overexploitation may occur and explores adjustments to model parameters to prevent such a scenario where feasible. The findings are derived through both analytical methods and visual representation via bifurcation diagrams to illustrate the dynamics' behavior.

The paper presents a comprehensive and insightful analysis of a Common-Pool Resource Game with several noteworthy attributes that contribute to its positive perception.

            The research addresses an important real-world issue concerning resource management and exploitation. This relevance is essential in the field of economics, as it can provide valuable insights into strategies for sustainable resource use.

            The paper delves into the dynamics of the game, particularly focusing on the behavior of fixed points, including the Nash Equilibrium, and the stability of the system. This theoretical depth adds to its academic significance.

The incorporation of Cournot-type payoff functions and the study of resource overexploitation provide a bridge between economic theory and environmental concerns. This interdisciplinary approach is both innovative and timely.

The paper employs a combination of analytical and graphical methods, including bifurcation diagrams. This comprehensive approach makes it accessible to a wider audience and enhances the clarity of the findings.

By investigating how parameter adjustments can prevent resource overexploitation, the research offers actionable insights that can inform policy decisions. This practical application contributes to its societal impact.

In summary, the paper offers a rigorous and multifaceted examination of a pressing economic and environmental issue. Its relevance, depth, interdisciplinary approach, analytical methods, and potential policy implications make it a valuable contribution to the field of economic theory and resource management.

I have only to note:

-Regarding equations very often something is missing on the left-hand side in front of the bracket or for what are the brackets good?

>> The brackets have been removed in the equations in which did not correspond.

-The English can be slightly improved. I propose to proofread the paper again, at best by someone who has good English skills

>> Thank you very much for your comment. We have reviewed the paper to improve our English.

Round 2

Reviewer 1 Report

Comments and Suggestions for Authors

The authors have taken into account my previous comments, and the manuscript is in a much better shape now. I'd like to suggest an accept decision.